# Structure of a DNA polymerase abortive complex with the 8OG:dA base pair at the primer terminus

Vinod K. Batra[1] & Samuel H. Wilson[1✉]

Adenine frequently pairs with the Hoogsteen edge of an oxidized guanine base (8OG) causing G to T transversions. The (*syn*)8OG:dA base pair is indistinguishable from the cognant base pair and can be extended by DNA polymerases with reduced efficiency. To examine the structural basis of this reduced efficiency, we sought to obtain the structure of the "product" complex of DNA polymerase (pol) β with the (*syn*)8OG:dA base pair at the primer terminus by soaking the binary complex crystals with a hydrolysable dCTP analogue complementary to the template base G. Crystallographic refinement of the structure revealed that the adenine of the (*syn*)8OG:dA base pair had been expelled from the primer terminus and a dCMP was inserted opposite 8OG in a reverse orientation; another uninserted molecule of the analogue was bound to the templating base G. This leads to an abortive complex that could form the basis of oxidatively-induced pol β stalling.

[1] Genome Integrity and Structural Biology Laboratory, National Institute of Environmental Health Sciences, National Institutes of Health, 111 T.W. Alexander Drive, Research Triangle Park, NC 27709, USA. ✉email: wilson5@niehs.nih.gov

Genomic DNA is under constant attack from reactive oxygen species (ROS) generated internally by endogenous chemical processes and through external sources. Oxygen free radicals cause oxidation of guanosine leading to the production of 8-oxodGTP. The mutagenic potential of 8-oxodGTP has been widely investigated and the results have been extensively reviewed[1,2]. Most DNA polymerases insert 8-oxodGTP by virtue of Hoogsteen base pairing opposite the adenine base, and this insertion can be with a relatively high catalytic efficiency. OGG1-initiated base excision repair (BER) removes 8OG from genomic DNA preventing mutagenesis (Supplementary Fig. 1a). MUTYH-initiated Pol λ or Pol β dependent BER can excise adenine from DNA, as depicted (Supplementary Fig. 1b)[3]. If left unrepaired, the 8OG:dA base pair is readily extended by most polymerases leading to G to T transversions, as the 8OG:dA base pair is not recognized as a lesion by the mismatch repair machinery or exonucleases; this is because of resemblance of the 8OG:dA base pair hydrogen bonding to that of a dT:dA base pair.

## Results

**Overall similar global conformation of the structure.** Binary complex crystals with the 8OG:dA base pair at the primer terminus ($n-1$ position) were grown; the templating base in the single-nucleotide gapped DNA was the natural dG base These crystals were then soaked with a hydrolysable dCTP analog (dCMPP(CH$_2$)P) with the hope of obtaining a structure resembling the product structure. The rate of incorporation of dCMPP(CH$_2$)P is slower than that of natural dCTP because of the higher pKa[4,5]. The soaking experiments were conducted at concentrations which are several fold higher than the in vivo physiological concentration of the analog to obtain near 100% occupancy of the analog in the crystals because of its slower rate of diffusion in the crystals. This property of "in crystallo" slower rate of incorporation of substrate can provide additional information(s) about the reaction intermediates/transition state. Diffraction quality data at 2.0 Å resolution were collected on these crystals (Table 1). The global conformation of the enzyme was similar to that of a ternary complex structure with normal substrates (3PDB ID: 3RJH[6]) with rmsd of 0.295 Å (Supplementary Fig. 2a).

**Unexpected features of the structure.** However, during structure refinement, several surprising features were revealed (Fig. 1a), and the structure appears to represent an abortive ternary complex: The 8OG:dA base pair at the template:primer terminus is disrupted; the primer strand is extended by one nucleotide (with dCMP); 8OG in the templating strand had moved from the *syn* to *anti* conformation and the newly inserted dCMP forms a wobble base pair with the *anti* 8OG, but with a "reversed O3′ polarity" caused by the N-glycosidic bond around N1 of the base and the C1′ of the sugar rotation approximately 210° so that the O3′ positions itself in the major groove. Reversal of the O3′ terminal polarity has also been observed in several primer terminus mismatch ternary complexes of DNA polymerase β[7]. (Fig. 1b–d).

**Stabilization of the extra-helical base.** The original primer base dA adopted an extra-helical conformation (Fig. 1c) stabilized by stacking interactions with Arg40 and Asp276 (Supplementary Fig. 2b). Arg40 forms a salt-bridge with Asp276 when the lyase domain transitions from an open binary complex to closed ternary complex upon nucleotide binding. This has consistently been found in pol β structures.

**Formation of the ternary complex by the binding of an additional analog molecule.** A second dCTP molecule is bound in the active site opposite the natural dG base in the templating strand

**Table 1 Crystallographic data collection and refinement statistics.**

| | |
|---|---|
| Data collection | |
| Space Group | P2$_1$ |
| Cell dimensions | |
| *a, b, c* (Å) | 50.8 80.5 55.6 |
| *α, β, γ* (°) | 90.00 107.8 90.00 |
| Resolution (Å) | 2.07 (2.00)[a] |
| $R_{sym}$ or $R_{merge}$ | 0.082 (0.342)[a] |
| $I/\sigma I$ | 12.9 (2.7)[a] |
| Completeness (%) | 97.3 (97.0)[a] |
| Redundancy | 2.8 (2.5)[a] |
| Refinement | |
| Resolution (Å) | 2.00 |
| No. reflections | 28010 |
| $R_{work}/R_{free}$ | 19.3/23.8 |
| No. of atoms | 3742 |
| Protein | 2608 |
| Ligand/ion | 688 |
| Water | 446 |
| B-factors | |
| Protein | 21.61 |
| Ligand/ion | 29.01 |
| Water | 29.37 |
| R.M.S. deviations | |
| Bond lengths (Å) | 0.008 |
| Bond angles (°) | 0.936 |

[a]Values in parentheses are for the highest-resolution shell.

(Fig. 1d). Although this dCMPP(CH$_2$)P molecule is bound in the normal fashion, the reverse polarity of the primer terminus O3′, renders the O3′-Pα distance (12.7 Å) incompatible for catalysis.

**Postulated events leading to the formation of abortive ternary complex.** Postulated steps in formation of this abortive ternary complex structure (Fig. 1a) are summarized in Fig. 2. In the first step, the primer 8OG:dA base pair is extended by dCMP insertion opposite dG on a one-nucleotide gapped substrate. Next, the 8OG–dG dinucleotide in the template strand and newly formed dA–dC dinucleotide in the primer strand undergo slippage. This promotes disruption of the 8OG:dA base pair, along with extrusion of the dA base. Realignment of the dC base in a wobble conformation with 8OG occurs, but with a reverse orientation. Subsequently, another molecule of dCMPP(CH$_2$)P binds opposite the template strand dG base, stabilizing the abortive ternary complex.

## Discussion

The stalled DNA synthesis complex at different steps, as described in Fig. 2, will have different biological consequences most likely in a 8OG:G sequence context. For example, the abortive ternary complex may block BER leading to adverse consequences. The polarity of the wobble base pair may reverse to attack the Pα of the incoming dCTP leading to an insertion mutation. Still another possibility is that the 8OG:dC (wobble) base pair is recognized as a mismatch structure (Supplementary Fig. 2c), and this is consistent with an induced fit mechanism where a mismatched primer terminus is misaligned relative to the correct incoming nucleotide, deterring further extension[7].

The structure reported here supports the notion that strand slippage is a fundamental structural mechanism by which abortive complexes can be formed when a polymerase encounters an oxidatively induced 8OG lesions. While the results described illustrate the capacity of pol β and the substrate system to form

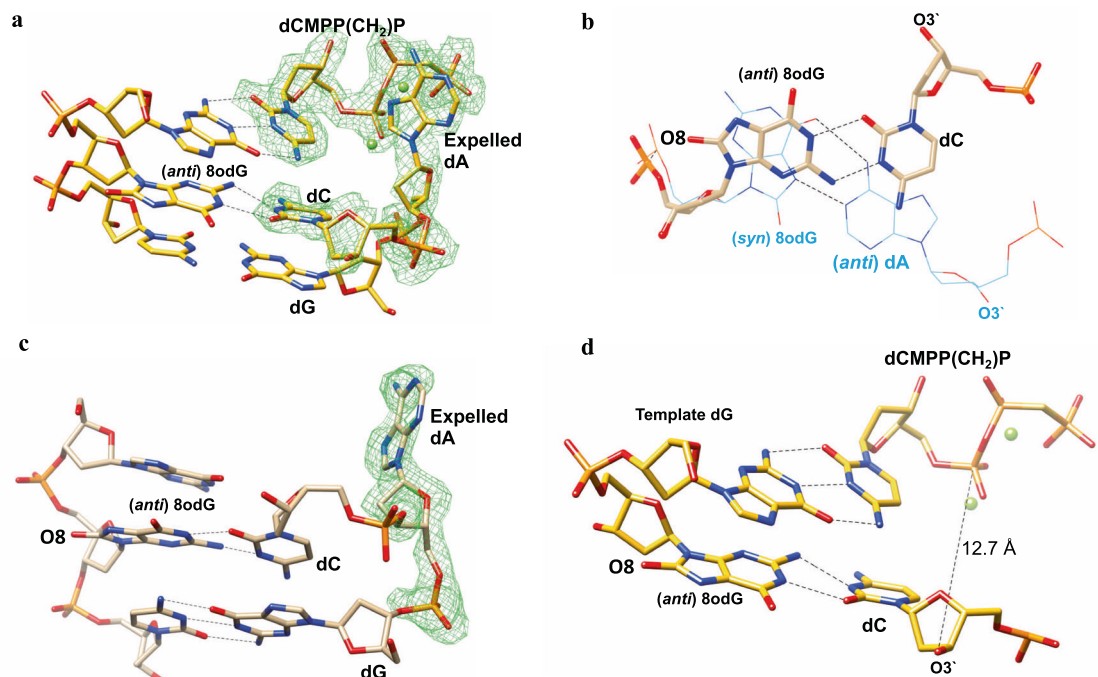

**Fig. 1 Salient features of the structure. a** Structure of the active site of DNA polymerase β showing the abortive ternary complex formed through dynamics in the DNA polymerase β active site, as a function of the 8OG:dA base pair in the template (8OG):primer (dA) position of a single-nucleotide gap substrate DNA. A simulated-annealing omit map contoured at $3\sigma$ is shown in green for the extra-helical base, incoming dCMPP(CH₂)P and the newly inserted dCMP as the primer terminus. **b** Overlay of the 8OG:dA base pair from the current structure (gold) on a reference ternary complex structure (PDB ID 3RJH) shown as a wire representation (cyan). Surprisingly, the current structure shows 8OG in the *anti*-conformation forming a wobble base pair with newly inserted dCMP. The 8OG:dA base pair in the primer position of the starting binary complex had been disrupted. The reference ternary complex structure shows the syn conformation of 8OG pairing with the *anti*-conformation of dA. **c** A figure showing the extra-helical dA base that was in template-primer 8OG: dA base pair. A simulated-annealing omit map contoured at $3\sigma$ is shown in green for the extra-helical dA base. **d** A figure illustrating the additional bound dCMPP(CH₂)P molecule forming an abortive ternary complex in the active site.

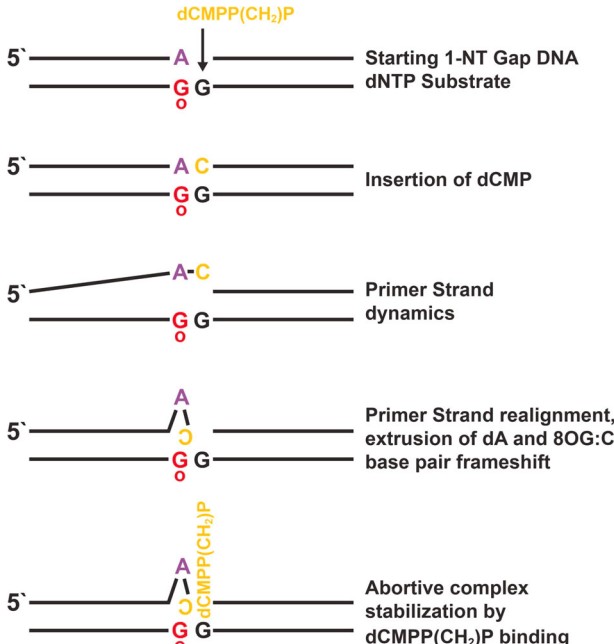

**Fig. 2 Postulated events involved in formation of the abortive ternary complex described here.** The bases have been color coded as depicted. $G_o$ denotes oxidized G (8OG).

and stabilize the abortive complex under the conditions used, more work will be required to understand the significance of this complex in vivo.

## Methods

**Crystallization of the Pol β substrate complex.** Human DNA polymerase β was overexpressed in *Escherichia coli* and purified[8]. The binary complex crystals with 8OG:dA at the primer terminus and a G as the templating base at the gap were grown as previously described[6]. The sequence of template with 8OG at $n-1$ position was as follows: 5′-CCG ACG GCG CAT CAG C-3′. The primer (10-mer) sequences were 5′-GCT GAT GCG A-3′. Annealing with primer created a 8OG:dA base pair at primer (terminus) after annealing. The templating base was dG. Downstream primer was phosphorylated and was 5′-GTC CC-3′. All the oligo-nucleotides were purchased from Integrated DNA Technologies, Inc. Binary complex crystals were grown in 50 mM imidazole, pH 7.5, 20% PEG3350, 90 mM sodium acetate. These crystals were then soaked in artificial mother liquor (50 mM Imidazole, pH 7.5, 20% PEG3350, 90 mM sodium acetate, 200 mM MgCl₂) with a hydrolysable analog (5 mM dCMPP(CH₂)P) and 12% ethylene glycol as cryoprotectant. Diffraction quality data were collected on the soaked crystal as described below.

**Data collection and structure determination.** Data were collected at 100 K on a CCD detector system mounted on a MiraMax®-007HF (Rigaku Corporation) rotating anode generator. Data were integrated and reduced with HKL2000 software[9]. Ternary substrate complex structures was determined by molecular replacement with a previously determined structure of pol β complexed with 8OG:dA base pair at the primer terminus and incoming dCMPCF₂PP (PDB ID 3RJH)[6].

**Reporting summary.** Further information on research design is available in the Nature Research Reporting Summary linked to this article.

## Data availability

The atomic coordinates and structure factors of the reported structure have been deposited in the RCSB Protein Data Bank (PDB) with the accession number 6W2M.

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

## Acknowledgements

This research was supported by the Intramural Research Program of the NIH, National Institute of Environmental Health Sciences project Nos. 1ZIAES050158 and 1ZIAES050161, (to S.H.W.). Molecular graphics images were produced using the Chimera package from the Resource for Biocomputing, Visualization, and Informatics at the University of California, San Francisco (supported by NIH P41 GM-103311).

## Author contributions

Conceptualization, methodology, formal analysis, and investigation, V.K.B.; Writing—original draft, V.K.B.; Writing—review and editing, V.K.B. and S.H.W.

## Competing interests

The authors declare no competing interests.
