## [Peer Review File · Communications Biology]

Reviewers' comments:

Reviewer #1 (Remarks to the Author):

Batra and Wilson provide structural evidence for a new intermediate in the base excision repair mechanism of pol beta. In particular, the authors were studying the structural basis of weak primer extension kinetics that occur after an adenosine nucleotide has been misincorporated opposite an oxidized guanosine nucleotide (8OG) in the template. The X-ray crystal structure of a single nucleotide addition reaction performed in crystallo produced a surprising abortive complex. Following nucleotide incorporation, which in this case is a dCTP analog opposite natural guanine, the primer-template duplex undergoes a frameshift pushing the adenosine nucleotide into an extra-helical position so that the dCMP adduct can base pair with 8OG. The findings are unusual and should be of broad scientific interest to the readers of Communications Biology.

Minor Comments

1. The legend of scheme 1 should include a color code and a description of the starting complex as a replication intermediate containing a one nucleotide gap.

2. The authors should be consistent and clear in the usage of nucleotide terminology and nomenclature in both the main text and SI:

- dCTPP(CH₂)P vs dCMPP(CH₂)P
- hydrolysable vs non-hydrolysable (found in SI)

3. The R_{work}/R_{free} values are missing from Supplementary Table 1.

4. Can the authors comment on the possibility that the abortive ternary complex may be driven by the presence of excess dCTP that is unique to their system and may not reflect physiological relevance? Is there experimental evidence of abortive ternary complexes?

Reviewer #2 (Remarks to the Author):

In the present communication, Batra and Wilson describe a structure of DNA polymerase β derived from the attempt to visualize the extension from an 8-oxodG:dA by dC. The dA at the primer end was extended by dCMP, but surprisingly, the nascent strand underwent a rearrangement with the terminal inserted dC being in wobble pairing with the 8-oxodG in anti-configuration and a new incoming dCTP analog pairing with the following dG at the template strand position +1. The complex is abortive, as the unusual positioning of the primer and in the complex precludes phosphodiester bond formation at the catalytic centre. Based on the analysis of the presented structure, the authors suggest a new structural mechanism by which DNA synthesis could be stalled at sites of oxidative damage in the template strand.

The manuscript is a short communication based on a single new crystal structure of human DNA polymerase β . The presented data are sound and the structure is rather intriguing. On the other hand, in the absence of additional support for the suggested stalling mechanism, one has to consider that the structure may represent an artefact induced by the crystallization conditions rather than a biologically relevant stalling complex. This is, because

- i. The concentration of the hydrolysable 5mM dCMPP(CH₂)P is 100-1000x higher than physiological dCTP concentrations in the cell,
- ii. After incorporation of a dCMP at the primer end, the DNA could not be extended any more. This is in contrast to the situation in the cell, where the multistep process of abortive complex formation depicted here would have to compete with the incorporation of the next nucleotide.

It is conceivable that the authors would not like to undertake an extensive kinetic analysis to

provide additional support for their proposed stalling model, and I believe that the presented structural data are interesting to the field. In my opinion, the manuscript should anyway discuss the following aspects:

1. The authors should briefly and critically embrace the possibility that the structure is favoured by the non-physiological experimental conditions during crystallization and soaking of the crystals (see issues raised above)
2. The authors should discuss in a little more detail the stabilization of the dA by Asp260 and Arg40. For instance, are these residues conserved in Pol β ? Is their position shifted compared to other structure of human Pol β ?
3. The authors should discuss, if the proposed mechanism would be conceivable for other sequence context, or is it a mechanism only possible in case of dG following the 8-oxo-dG?
4. The authors should provide support (e.g. by comparison with published structural or biochemical data), why they believe the observed abortive stalling mechanism could be utilised by structurally unrelated DNA polymerases, especially replicative DNA polymerases of family B.

Helmut Pospiech

Reviewer #3 (Remarks to the Author):

Dear editor,

In this work Batra and Wilson present a structure of human Pol β with a 8oxodG in the template strand and looped out base in the primer strand. It is a nice and brief manuscript, with a clear to-the-point structure description.

I only have a few comments & suggestions

1) The reason for the use of dCMPP(CH₂)P rather than regular dCTP is not clear. What is the benefit of using this analog? Does Pol β incorporate this nucleotide differently from dCTP?

2) Page 2, towards end of page: "newly inserted dCMP forms a wobble base pair with the anti 8OG, but with a "reverse O3' polarity". Please elaborate on "reverse O3' polarity", e.g. "where the ribose is rotated by $\langle i \rangle$ degrees from the canonical position".

3) Two lines below: "A second dCTP molecule is bound ..." replace with: " A dCMPP(CH₂)P molecule is bound..."

4) Page 3, second paragraph: "...the abortive complex may block replication..."

This is unlikely. This particular structure is observed in Pol β because two residues Arg40 & Asp276 support the flipped-out adenine base. As soon as Pol β releases the DNA, the primer strand will adjust itself to a more natural conformation of 8oxodG:dA and dG:dC at the -2 and -1 position, respectively. As 8oxodG:dA only minimally distorts the DNA helix, this substrate would not impose a block to DNA replication.

Instead, it is possible that the abortive Pol β -8oxodG structure is more stable than Pol β bound to canonical DNA, and that it stalls DNA synthesis long enough to recruit DNA repair enzymes to the lesion.

Reviewers' comments:

Reviewer #1 (Remarks to the Author):

Batra and Wilson provide structural evidence for a new intermediate in the base excision repair mechanism of pol beta. In particular, the authors were studying the structural basis of weak primer extension kinetics that occur after an adenosine nucleotide has been misincorporated opposite an oxidized guanosine nucleotide (8OG) in the template. The X-ray crystal structure of a single nucleotide addition reaction performed in crystallo produced a surprising abortive complex. Following nucleotide incorporation, which in this case is a dCTP analog opposite natural guanine, the primer-template duplex undergoes a frameshift pushing the adenosine nucleotide into an extra-helical position so that the dCMP adduct can base pair with 8OG. The findings are unusual and should be of broad scientific interest to the readers of Communications Biology.

Minor Comments

Comment No.	Reviewers' comments	Author's reply
1	The legend of scheme 1 should include a color code and a description of the starting complex as a replication intermediate containing a one nucleotide gap.	The legend of Scheme 1 has been modified to indicate the starting complex as a one nucleotide gap structure and a color code is included on line 250.
2	The authors should be consistent and clear in the usage of nucleotide terminology and nomenclature in both the main text and SI: • dCTPP(CH₂)P vs dCMPP(CH₂)P• hydrolysable vs non-hydrolysable (found in SI)	dCTPP(CH ₂)P has been changed to dCMPP(CH ₂)P on line 121. non-hydrolysable should have read hydrolysable and has been corrected on line 108.
3	The Rwork/Rfree values are missing from Supplementary Table 1.	The missing Rwork/Rfree values have been filled in Table 1.
4	Can the authors comment on the possibility that the abortive ternary complex may be driven by the presence of excess dCTP that is unique to their system and may not reflect physiological relevance? Is there experimental evidence of abortive ternary complexes?	The crystals are usually grown in high molecular weight polyethylene glycols which are highly viscous and thus permit only limited diffusion of the solutes. Therefore, the soaking experiments are conducted at a relatively high concentrations of the substrate/analog to obtain near 100% occupancy of the compounds in the crystals. Crystallography conditions are often different from those expected in vivo . Nevertheless, useful information is gained through crystallography. The sentences describing this have been included on lines

Reviewer #2 (Remarks to the Author):

In the present communication, Batra and Wilson describe a structure of DNA polymerase β derived from the attempt to visualize the extension from an 8-oxodG:dA by dC. The dA at the primer end was extended by dCMP, but surprisingly, the nascent strand underwent a rearrangement with the terminal inserted dC being in wobble pairing with the 8-oxodG in anti-configuration and a new incoming dCTP analog pairing with the following dG at the template strand position +1. The complex is abortive, as the unusual positioning of the primer and in the complex precludes phosphodiester bond formation at the catalytic centre. Based on the analysis of the presented structure, the authors suggest a new structural mechanism by which DNA synthesis could be stalled at sites of oxidative damage in the template strand.

The manuscript is a short communication based on a single new crystal structure of human DNA polymerase β . The presented data are sound and the structure is rather intriguing. On the other hand, in the absence of additional support for the suggested stalling mechanism, one has to consider that the structure may represent an artefact induced by the crystallization conditions rather than a biologically relevant stalling complex. This is, because

- i. The concentration of the hydrolysable 5mM dCMPP(CH₂)P is 100-1000x higher than physiological dCTP concentrations in the cell,
- ii. After incorporation of a dCMP at the primer end, the DNA could not be extended any more. This is in contrast to the situation in the cell, where the multistep process of abortive complex formation depicted here would have to compete with the incorporation of the next nucleotide.

It is conceivable that the authors would not like to undertake an extensive kinetic analysis to provide additional support for their proposed stalling model, and I believe that the presented structural data are interesting to the field. In my opinion, the manuscript should anyway discuss the following aspects:

Comment No.	Reviewers' comments	Author's reply
1	The authors should briefly and critically embrace the possibility that the structure is favoured by the non-physiological experimental conditions during crystallization and soaking of the crystals (see issues raised above)	This point has been answered in replies to comment 4 of reviewer 1. and point 1 of reviewer 3.
2	The authors should discuss in a little more detail the stabilization of the dA by Asp260 and Arg40. For instance, are these residues conserved in Pol β ? Is their position shifted compared to other structure of human Pol β ?	Asp276 forms a salt-bridge with Arg40. These residues are conserved in other structures in this system. This has been incorporated in the revised text on lines 136 -143.

3	The authors should discuss, if the proposed mechanism would be conceivable for other sequence context, or is it a mechanism only possible in case of dG following the 8-oxo-dG?	It is likely that this effect would be more pronounced with a 8OG-G sequence context and has been added on line 164. We have not tested other sequences, but such experiments are under consideration.
4	The authors should provide support (e.g. by comparison with published structural or biochemical data), why they believe the observed abortive stalling mechanism could be utilised by structurally unrelated DNA polymerases, especially replicative DNA polymerases of family B.	We do not have evidence regarding replication per se or other polymerases. therefore, we have revised so as to limit the interpretation to the gap filling repair polymerase pol beta on lines 173-175.

Helmut Pospiech

Reviewer #3 (Remarks to the Author):

Dear editor,

In this work Batra and Wilson present a structure of human Pol β with a 8oxodG in the template strand and looped out base in the primer strand. It is a nice and brief manuscript, with a clear to-the-point structure description.

I only have a few comments & suggestions

Comment No.	Reviewers' comments	Author's reply
1	The reason for the use of dCMPP(CH ₂)P rather than regular dCTP is not clear. What is the benefit of using this analog? Does Pol β incorporate this nucleotide differently from dCTP?	As stated in the abstract, the aim of the study was to obtain a "product complex" structure not an "abortive complex". The rate of incorporation of dCMPP(CH ₂)P is slower than that of natural dCTP because of the higher pKa (10.5) (Oertel et al. Biochemistry, 2016). The property of 'in crystallo' slower rate of incorporation of substrate has been exploited to study the reaction intermediates/transition states by time-lapse crystallography. This statement has been added on lines 122-128.
2	towards end of page: "newly inserted dCMP forms a wobble base pair with	This sentence has been expanded to indicate that the N-glycosidic bond around N1 of the

	the anti 8OG, but with a "reverse O3' polarity". Please elaborate on "reverse O3' polarity", e.g. "where the ribose is rotated by nnn degrees from the canonical position".	base and C1' of the sugar rotates approximately 210° so that the O3' positions itself in the major groove. Reversal of the O3' terminal polarity has also been observed in several primer terminus mismatch ternary complexes of DNA polymerase β (Batra et al. Structure, 2016) on lines 136-138.
3	Two lines below: "A second dCTP molecule is bound ..." replace with: "A dCMPP(CH₂)P molecule is bound..."	"A second dCTP molecule is bound ..." has been replaced with: "A dCMPP(CH₂)P molecule is bound..." on line 143."
4	Page 3, second paragraph: "...the abortive complex may block replication..." This is unlikely. This particular structure is observed in Pol β because two residues Arg40 & Asp276 support the flipped-out adenine base. As soon as Pol β releases the DNA, the primer strand will adjust itself to a more natural conformation of 8oxodG:dA and dG:dC at the -2 and -1 position, respectively. As 8oxodG:dA only minimally distorts the DNA helix, this substrate would not impose a block to DNA replication. Instead, it is possible that the abortive Pol β-8oxodG structure is more stable than Pol β bound to canonical DNA, and that it stalls DNA synthesis long enough to recruit DNA repair enzymes to the lesion.	Yes, we agree entirely with the reviewer. The use of the word "replication" was intended to simply mean DNA synthesis. We now appreciate the problem with this, and the term has been corrected in the revised manuscript on line 166. We appreciate the assistance of the reviewer on this point.

REVIEWERS' COMMENTS:

Reviewer #1 (Remarks to the Author):

The authors have addressed all of my concerns.

Reviewer #2 (Remarks to the Author):

In the revised manuscript, the authors have addressed the issues raised by the initial version of the manuscript. I therefore can recommend acceptance.

Reviewer #3 (Remarks to the Author):

I am happy with the revised manuscript. The authors have adequately addressed all the points raised by the reviewers, and resolved any unclarities in the manuscript.